# Factors Associated with an Increase in On-Site Time of Pediatric Trauma Patients in a Prehospital Setting: A Nationwide Observational Study in Japan

**DOI:** 10.3390/children9111658

**Published:** 2022-10-29

**Authors:** Shunichi Otaka, Hiroyuki Ohbe, Ryuhei Igeta, Takuyo Chiba, Shunya Ikeda, Takashi Shiga

**Affiliations:** 1Department of Emergency Medicine, International University of Health and Welfare Narita Hospital, Chiba 286-8520, Japan; 2Graduate School of Medicine, International University of Health and Welfare, Chiba 324-8501, Japan; 3Department of Clinical Epidemiology and Health Economics, School of Public Health, The University of Tokyo, Tokyo 113-8654, Japan; 4Department of Public Health, School of Medicine, International University of Health and Welfare, Chiba 324-8501, Japan

**Keywords:** database, multiple imputation, on-site time, pediatric, prehospital, trauma

## Abstract

The factors that prolong the on-site time in pediatric trauma cases in a prehospital setting are unknown. We investigated these factors using a national trauma registry in Japan. We identified pediatric trauma patients aged ≤18 years, from January 2004 to May 2019. We categorized cases into shorter (≤13 min) and longer (>13 min) prehospital on-site time groups. We performed multivariable logistic regression analysis with multiple imputations to assess the factors associated with longer prehospital on-site time. Overall, 14,535 patients qualified for inclusion. The median prehospital on-site time was 13 min. In the multivariable logistic regression analysis, the longer prehospital on-site time was associated with higher age; suicide (Odds ratio [OR] 1.27; 95% confidence interval [CI] 1.03–1.57); violence (OR 1.74; 95%CI 1.27–2.38); higher revised trauma score, abbreviated injury scale > 3 in the spine (OR 1.25; 95%CI 1.04–1.50), upper extremity (OR 1.26; 95%CI 1.11–1.44), and lower extremity (OR 1.25; 95%CI 1.14–1.37); immobilization (OR 1.16; 95%CI 1.06–1.27); and comorbid mental retardation (OR 1.56; 95%CI 1.11–2.18). In light of these factors, time in the field could be reduced by having more pediatric emergency physicians and orthopedic surgeons available.

## 1. Introduction

It has been reported that the time course from the onset of trauma to hospital arrival is an important factor in the prognosis of trauma patients. Several previous studies have suggested that a delay in prehospital time may affect prognosis in critically ill trauma patients [1,2,3,4,5].

Prehospital time includes the time from the emergency call to arrival of the ambulance on site, the time required for treatment and care on-site, the time for selection of a hospital, and the time from leaving the site to arriving at the hospital. Efforts have been made to shorten the prehospital time from the time of the call to arrival at the scene and the time from leaving the site to arrival at the hospital by organizing the transportation network and using advanced transportation involving helicopters [6,7]. On the other hand, time spent on site (on-site time) includes time for medical treatment of the patient and selection of a hospital to which to transport the patient. Regarding the difference in treatment based on the severity and urgency of the patient, it is a matter of prioritizing adequate treatment at the scene or “scoop and run” [8]. However, other unnecessary time spent in the field with critically ill patients may affect their outcomes [9]. Even for non-severe trauma patients, prolonged on-site time could be a public health issue, because it delays dispatch to the next emergency call. However, the factors that prolong on-site stay, other than the severity of the trauma and the time required to stay to perform a strategic procedure, are not known.

A previous study identified male sex, the severity of the injury, and injury during nighttime, holidays, and weekends as factors that may increase difficulty in selecting a hospital, requiring four or more phone calls at on-site [10]. However, the previous study did not include information on the patient’s medical history and did not adequately adjust for the severity of trauma. Furthermore, that study excluded pediatric patients.

In the present study, we sought to identify factors that contribute to prolonged on-site time for pediatric patients presenting to the hospital by ambulance, including factors associated with increased length of stay at the scene, using a national trauma registry in Japan.

## 2. Methods

### 2.1. Study Design and Japan Trauma Data Bank

This was a retrospective cohort study using a national trauma registry in Japan, called the Japan Trauma Data Bank. The registry is described in detail elsewhere [11,12,13]. Briefly, the registry is managed by the Japanese Association for the Surgery of Trauma (Trauma Surgery Committee) and the Japanese Association for Acute Medicine (Committee for Clinical Care Evaluation). This registry is similar to the trauma registries in North America, Europe, and Oceania [14]. By 2021, 292 major hospitals had registered cases in this registry [15]. The data were collected from participating hospitals via the Internet. Mainly patients with Abbreviated Injury Scale (AIS) ≥ 3 were registered. The physicians or technicians were required to complete an AIS coding course to register the data [16]. The data were collected from January 2004 and are still accumulating. The registry contains the following patient data: age (years); sex; comorbidities: dates of the trauma, call time to Emergency Medical Services (EMS), type of trauma, cause of the trauma, prehospital vital signs, procedures by EMS; time series data on pre-hospital activities, and trauma severity using the Revised Trauma Score (RTS) [17,18]; AIS version 1998 [19]; Injury Severity Score (ISS) [20]; and Trauma Injury Severity Score (TRISS) [21,22], interventions before hospital arrival, and comorbidities.

### 2.2. Study Setting and Emergency Medical Service System in Japan

In 2019, there were 726 fire stations with dispatch centers. The EMS system is operated by the Fire and Disaster Management Agency of Japan. All patients are transported to a hospital, except for cases of decapitation, incineration, decomposition, and rigor mortis cases. In a prehospital setting, the EMS team can perform the following procedures for trauma patients: provide oxygen; immobilize a patient with a cervical collar or backboard; perform chest compressions; and use an automated external defibrillator. Under remote medical direction, for trauma patients in shock or in cardiopulmonary arrest, the EMS team can perform the following procedures: secure a peripheral intravenous line; administer Ringer’s lactate solution; administer adrenaline (epinephrine) intravenously; and establish an advanced airway with an endotracheal tube, laryngeal mask airway, Combi-tube, and esophageal gastric tube airway. In the Japanese EMS system, requests for acceptance made by the EMS team are sometimes denied by the hospitals, which may contribute to prolonging the on-site time.

### 2.3. Patient Selection

Using the Japan Trauma Data Bank, we collected data on pediatric trauma patients aged ≤18 years from 1 January 2004, to 28 May 2019. We excluded patients with an on-site time of more than 120 min as reflecting unrealistic time records.

### 2.4. Outcomes

We defined prehospital on-site time as the time between the EMS team arrival at the scene and their departure from the scene. We calculated the median on-site time and divided the patients into two groups: patients with shorter on-site time (“shorter on-site time” group) and patients with longer on-site time (“longer on-site time” group). We compared the variables between the groups and investigated the factors that contributed to prolonging the on-site time.

### 2.5. Variables

We considered the following variables: age; sex; year; month; days of the week; call time; type of trauma (traffic accident, fall, other blunt trauma, penetrating trauma), cause of trauma (accident, occupational injury, suicide, violence, others, unknown); vital signs on site (degree of consciousness, pulse rate, systolic blood pressure, respiratory rate, cardiopulmonary arrest); RTS; AIS score; interventions before hospital arrival (oxygen administration, immobilization (cervical collar and/or backboard immobilization), chest compression, intravenous line placement, defibrillation, and intubation); and comorbidities (mental disease, mental retardation, and physical disease, including hemodialysis dependent chronic kidney disease, malignancy, diabetes mellitus, ischemic heart disease, and cerebrovascular disease).

We divided patients’ age into five categories: 0, 1–3, 4–6, 7–12, and 13–18; the year of the case into five categories: 2004–2007, 2008–2010, 2011–2013, 2014–2016, and 2017–2019; the month of the injury into four categories: January–March, April–June, July–September, and October–December; the time of the injury into daytime (09:00–16:59) and nighttime (17:00–08:59), according to previous studies [10,23,24]. The level of consciousness was evaluated by using the Japan Coma Scale (JCS) score [25,26,27], which is widely used as a standard in Japan and which correlates well with the Glasgow coma scale. The JCS score was subdivided into four categories: 0, alert and conscious; 1–3, drowsy but awake without stimulation; 10–30, lethargic and drowsy but can be aroused with stimulation; 100–300, coma. We regarded the following patients as having hypotension: children younger than 12 months with systolic blood pressure of < 50 mm Hg; children aged 1–5 years with systolic blood pressure of <60 mm Hg, and children older than 5 years with systolic blood pressure <70 mm Hg [28]. ISS was subgrouped by ≥16 points or <16 [20,29]. AIS was also subgrouped as ≥3 points or <3 points [30,31,32].

### 2.6. Statistical Analysis

First, we performed a univariate analysis using chi-square tests and *t*-tests to compare the variables between the shorter on-site time group and the longer on-site group. Then, we performed a multivariable logistic regression analysis for longer on-site time with the following variables: patient age; patient sex; year, month, and day of the week on which injury occurred; time of day that the injury occurred; type of trauma; cause of trauma; the presence of hypotension; RTS; AIS1 (head) ≥3; AIS2 (face) ≥3; AIS3 (neck) ≥3; AIS4 (thorax) ≥3; AIS5 (abdomen/pelvic content) ≥3; AIS6 (spine) ≥3; AIS7 (upper extremities) ≥3; AIS8 (lower extremities) ≥3; AIS9 (external, burns/other trauma) ≥3; interventions before hospital arrival (oxygen administration; immobilization; chest compression; intravenous line placement; defibrillation; and intubation); and comorbidities (mental disease, mental retardation, physical disease). Because data on some variables (sex, year, month, days of the week, time of day, type of trauma, cause of trauma, presence of hypotension, RTS, and intubation) were missing, we performed multiple imputations to account for bias caused by missing data [33,34]. In this study, we replaced each missing value with a set of substituted plausible values by generating 20 filled-in complete datasets using multiple imputations with the chained equation method and 20 iterations on the assumption of data missing at random [35]. All variables used in this multivariate analysis were included in the data imputation process. We computed odds ratios (ORs) and 95% confidence intervals (CIs) using Rubin’s rules [36].

For sensitivity analysis, we performed a complete case analysis excluding all patients with missing data.

Continuous variables are expressed as the median and interquartile range, and categorical variables are expressed as frequencies and percentages. A two-sided *p*-value of < 0.05 was considered statistically significant. All statistical analyses were performed with the general statistical package STATA (version 16.0; StataCorp, College Station, TX, USA).

### 2.7. Ethics

This study was approved by the Institutional Review Board of the International University of Health and Welfare, Narita Hospital (approval number: 22-Im-008 28 June 2022). The requirement for informed consent was waived due to the use of anonymized data.

## 3. Results

The patient flowchart is shown in Figure 1. There were 19,374 registered pediatric trauma patients aged ≤18 years. After adopting the exclusion criteria, 14,535 patients qualified for this study.

The baseline characteristics of all patients are shown in Table 1. The patient’s median age was 13 years. Therefore, we categorized cases into shorter (≤13 min) and longer (>13 min) prehospital on-site time groups. The median on-site time was 13 min (interquartile range 9–18 min). A total of 8053 patients were assigned to the shorter on-site time group and 6482 to the longer on-site time group.

The results of the univariate analysis showed that patients in the longer on-site time group were older, suffered injury at a later date, at nighttime, in a traffic accident, had more alert consciousness, less frequently had hypotension and cardiopulmonary arrest, had injury with less severity (higher RTS, lower ISS, and higher TRISS), required less oxygen administration and chest compression, more frequently required immobilization, and more often had a history of mental retardation than did patients in the shorter-prehospital-stay group (Table 2).

Table 3 shows a multivariable logistic regression analysis with multiple imputations for missing data. Older age was associated with higher ORs for longer on-site time. The more recent the injury, the higher was the OR for longer on-site time. Calls during daytime showed a higher proportion of shorter on-site time than did those received nighttime (OR 0.89; 95%CI 0.83–0.96; *p* = 0.002). Falls and other blunt trauma were associated with shorter on-site time, using traffic accident as the reference (OR 0.80; 95%CI 0.73–0.88; *p* < 0.001 and OR 0.69; 95%CI 0.61–0.78, *p* < 0.001, respectively). In terms of the cause of trauma, suicide and violence were associated with longer on-site time, using accident as a reference (OR 1.27; 95%CI 1.03–1.57; *p* = 0.03 and OR 1.74; 95%CI 1.27–2.38, *p* = 0.001, respectively). Higher RTS was associated with longer on-site time (OR 1.25; 95%CI 1.19–1.31; *p* < 0.001). AIS1 (head) ≥3 (OR 0.77; 95%CI 0.71–0.84; *p* < 0.001), AIS4 (thorax) ≥3 (OR 0.84; 95%CI 0.76–0.92; *p* < 0.001), and AIS5 (abdomen/pelvic content) ≥3 (OR 0.84; 95%CI 0.72–0.99; *p* < 0.04) were associated with shorter on-site time. AIS6 (spine) ≥3 (OR 1.25; 95%CI 1.04–1.50; *p* = 0.02), AIS7 (upper extremities) ≥3 (OR 1.26; 95%CI 1.11–1.44; *p* < 0.001), and AIS8 (lower extremities) ≥3 (OR 1.25; 95%CI 1.14–1.37; *p* < 0.001) were associated with longer on-site time. As for interventions before hospital arrival, oxygen administration (OR 0.85; 95%CI 0.78–0.93; *p* < 0.001) was associated with shorter on-site time. However, immobilization was associated with longer on-site time (OR 1.16; 95%CI 1.06–1.27; *p* = 0.001). Comorbid mental retardation in the injured patient was associated with longer on-site time as compared to patients without this comorbidity (OR 1.56; 95%CI 1.11–2.18; *p* = 0.009). Other mental disease and physical disease were not associated with longer on-site time.

The results of the complete case analysis are shown in Table 4. There were 4935 patients in the shorter on-site time group and 4593 patients in the longer on-site time group. Because only a small number of patients were treated with defibrillation, this factor was omitted from the analysis. The results were similar to those of the main analyses.

## 4. Discussion

We here compared factors that differed between pediatric trauma patients with longer on-site time and shorter on-site time using a national trauma registry in Japan. According to multivariate analysis with multiple imputations, the proportions of older pediatric patients; more recent injuries; calls at nighttime; injuries due to traffic accident, suicide, and violence; higher RTSs, cases with AIS6 (spine), AIS7 (upper extremities), and AIS8 (lower extremities) ≥ 3; use of immobilization; and patients with mental retardation were higher in the longer on-site time group than in the shorter-on-site time group. However, there were no differences between the groups in terms of physical disease. Compared to previous studies that focused on adult trauma patients, the severity of trauma was inconsistent. The factors of more recent injuries; injuries caused by suicide and violence; AIS6 (spine), AIS7 (upper extremities), and AIS8 (lower extremities) ≥ 3, immobilization; and patients with mental retardation were newly identified as factors prolonging on-site time.

There are several explanations for these results. First, our study focused on on-site time, whereas a previous related study focused on the difficulty of finding a hospital to which to transport the patients [10]. Second, injury in more recent years was a factor that prolonged prehospital stay. This may be because the number of emergency patients is increasing every year [37]. Consequently, emergency services may be overcrowded, making it difficult to accept emergency patients. Third, suicide and violent trauma are not only medical problems, but are complex social issues, which can make it difficult for hospitals to accept patients. Fourth, trauma to the spine or extremities is often less urgent, but is important in terms of functional prognosis, and require more specialized care in a hospital than trauma to the head or torso. In Japan, a hospital where definitive treatment can be completed needs to be selected as the first destination, which is difficult and increases the time spent in the field. Fifth, mental retardation was a factor associated with longer on-site time. Hospitals in Japan often provide both psychiatric and physical care. Compared to patients with those comorbidities, it is rare for patients with mental retardation alone to have a family doctor whom he or she sees regularly. Physicians may be hesitant to accept new patients with mental retardation. As a result, it is difficult to find a hospital to which to transport such patients, which prolongs the on-site time. In addition, communication with the patient may be difficult, which hampers obtaining the patient’s medical history on-site and which prolongs the length of on-site time.

Our results have several implications for shortening the on-site time for attending to pediatric trauma cases. To shorten the time spent in the prehospital field regardless of the patient’s social background or the location of the trauma, we believe that hospitals should play two types of roles. First, there should be a hospital that can provide an accurate diagnosis, appropriate life-saving treatment, and stabilization of the patient’s condition. Second, there should be a hospital that can handle patients who require surgery for functional prognosis or special management due to the patient’s background. With cooperation between such hospitals, smooth transfer of patients from one hospital to the next will facilitate the rapid admission of patients. To establish this system, there is a need for pediatric emergency physicians in the former type of hospital. Some countries, such as the United States, have a pediatric emergency medicine specialist system, but as yet, in 2022, such a system does not exist in Japan [38]. The number of emergency medicine specialists in Japan in 2020 is 4700, for a population of 125 million (14.7 million aged 0–14 years), or 3.7 emergency medicine specialists per 100,000 individuals. In the United States, on the other hand, there are 48,835 emergency medicine specialists for a population of 331 million (62.0 million aged 0–14 years), or 14.8 emergency medicine specialists per 100,000 individuals. [39,40]. The current number of pediatric specialists in the United States has not been surveyed. In 2018, the number of certified pediatric specialists was 3132 in the United States, as compared to 426 in Japan [41,42]. The number of pediatricians and emergency physicians in Japan remains very low, and it can be surmised that the number of physicians who can deal with pediatric trauma in acute phase is also very low. Second, pediatric orthopedic surgeons or orthopedic trauma surgeons are needed to deal with traumatic injuries to the limbs and spine [43,44]. Although these injuries are not directly involved in patient mortality, they are important for functional prognosis. Japan also has a shortage of such physicians; yet they are crucial for establishing a system that will shorten the time spent in the prehospital field.

There were several strengths in this study. First, to the best of our knowledge, no previous large study has investigated the factors that prolong the time spent on-site in pediatric trauma cases. Second, the nationwide database used here includes a large number of patients. Therefore, we were able to adjust for many factors in the multivariate analysis. Third, we adjusted for the severity of the trauma, whereas previous studies were markedly limited in that they did not account for trauma severity. Fourth, we conducted a multivariate analysis with multiple imputations and performed a complete case analysis as a sensitivity analysis, which confirmed the robustness of our analysis.

However, there were several limitations in this study. First, it was unclear whether the detailed reasons for staying longer on-site was due to the procedure, the selection of the hospital to which to transport the patient, or another reason. Second, in Japan, requests for acceptance by the EMS team are sometimes refused by hospitals, but in other countries and regions, patients can be transported as is; thus, our results may not apply to such situations. Third, due to the lack of local information in this database, we could not adjust for the characteristics of the local medical situation.

## 5. Conclusions

The proportions of older children; more recent injuries; calls during nighttime; injuries related to traffic accident, suicide, and violence; higher RTS; AIS6 (spine), AIS7 (upper extremities), and AIS8 (lower extremities) ≥3; the need for immobilization, and patients with mental retardation contributed to longer on-site time. Given the nature of these factors, the availability of more pediatric emergency physicians and orthopedic surgeons could expedite hospital admissions and reduce time in the field.

## Figures and Tables

**Figure 1 children-09-01658-f001:**
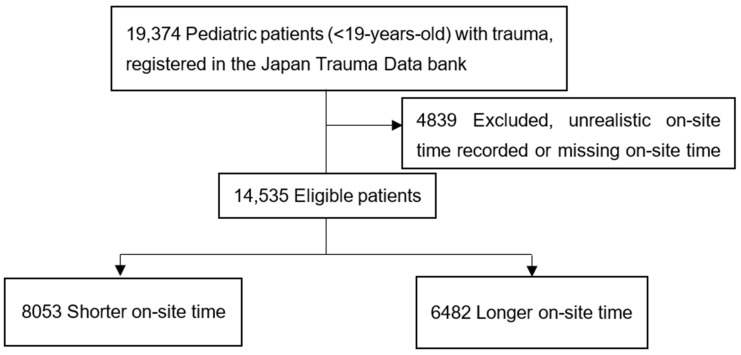
Patient flow chart.

**Table 1 children-09-01658-t001:** Characteristics of the study population.

Variables	Total (*n* = 14,535)	
Age: years, median (IQR)	13	(7–17)
Age groups: years, *n* (%)		
0	308	(2.1)
1–3	1241	(8.5)
4–6	1434	(9.7)
7–12	3859	(26.6)
13–18	7693	(52.9)
Sex, *n* (%)		
Male	10,485	(72.1)
Female	4046	(27.8)
Missing	4	(<0.1)
Year, *n* (%)		
2004–2007	1148	(7.9)
2008–2010	1856	(12.8)
2011–2013	3599	(24.8)
2014–2016	4926	(33.9)
2017–2019	2978	(20.5)
Missing	28	(0.2)
Month, *n* (%)		
January–March	2772	(19.1)
April–June	4167	(28.7)
July–September	4125	(28.4)
October–December	3443	(23.7)
Missing	28	(0.2)
Days of week, *n* (%)		
Weekdays	9316	(64.1)
Weekends/Holidays	5218	(35.9)
Missing	1	(<0.1)
Time, *n* (%)		
Daytime	6206	(42.7)
Nighttime	8239	(56.7)
Missing	90	(0.6)
Type of trauma, *n* (%)		
Traffic accident	8934	(61.5)
Fall	3575	(24.6)
Other blunt trauma	1717	(11.8)
Penetrating trauma	192	(1.3)
Missing	117	(0.8)
Cause of trauma, *n* (%)		
Accident	13,388	(92.1)
Occupational injury	83	(0.6)
Suicide	637	(4.4)
Violence	191	(1.3)
Others	78	(0.5)
Unknown/Missing	158	(1.1)
**Prehospital vital signs**		
Japan coma scale, *n* (%)		
0	6821	(46.9)
1–3	3766	(25.9)
10–30	1286	(8.9)
100–300	1873	(12.9)
Unknown/Missing	789	(5.4)
Pulse rate: bpm, median (IQR)	95	(80–110)
Missing, *n* (%)	1046	(7.2)
Systolic blood pressure: mmHg, median (IQR)	121	(110–135)
Hypotension	77	(0.5)
Missing, *n* (%)	2437	(16.8)
Respiratory rate, median (IQR)	24	(20–30)
Missing, *n* (%)	2136	(14.7)
Cardiopulmonary arrest	441	(3.0)
**Severity of trauma**		
RTS, median (IQR)	7.84	(7.55–7.84)
Missing, *n* (%)	2265	(15.6)
AIS		
AIS1 ≥ 3, *n* (%)	4552	(31.3)
AIS2 ≥ 3, *n* (%)	127	(0.9)
AIS3 ≥ 3, *n* (%)	31	(0.2)
AIS4 ≥ 3, *n* (%)	2721	(18.7)
AIS5 ≥ 3, *n* (%)	742	(5.1)
AIS6 ≥ 3, *n* (%)	557	(3.8)
AIS7 ≥ 3, *n* (%)	1109	(7.6)
AIS8 ≥ 3, *n* (%)	2331	(16.0)
AIS9 ≥ 3, *n* (%)	10	(0.1)
ISS, median (IQR)	10	(5–17)
ISS > 15	4692	(32.3)
ISS < 16	8737	(60.1)
Missing, *n* (%)	1106	(7.6)
TRISS, median (IQR)	0.99	(0.98–1.00)
Missing, *n* (%)	3333	(22.9)
**Interventions before hospital arrival**		
Oxygen	6933	(47.7)
Immobilization	8740	(60.1)
Chest compression	378	(2.6)
Intravenous line placement	156	(1.1)
Defibrillation	18	(0.1)
Intubation	67	(0.5)
Missing	826	(5.7)
**Comorbidities**		
Mental disease	272	(1.9)
Mental retardation	149	(1.0)
Physical disease	44	(0.3)
**On-site time, median (IQR)**	13	(9–18)

AIS, Abbreviated Injury Scale; IQR, Interquartile range; ISS, Injury Severity Score; RTS, Revised Trauma Score; TRISS, Trauma Injury Severity Score.

**Table 2 children-09-01658-t002:** Univariate analysis of factors associated with the length of on-site time.

Univariate Analysis					
Variables	Shorter On-Site Time (*n* = 8053)		Longer On-Site Time (*n* = 6482)		*p*-Value
Age: years, median (IQR)	12	(7–16)	14	(9–17)	<0.001
Age groups: years, *n* (%)					<0.001
0	213	(2.6)	95	(1.5)	
1–3	868	(10.8)	373	(5.8)	
4–6	916	(11.4)	518	(8.0)	
7–12	2171	(27.0)	1688	(26.0)	
13–18	3885	(48.2)	3808	(58.7)	
Sex, *n* (%)					0.07
Male	2224	(27.6)	1822	(28.1)	
Female	5829	(72.4)	4656	(71.8)	
Missing	0	(0.0)	4	(0.1)	
Year, *n* (%)					<0.001
2004–2007	740	(9.2)	408	(6.3)	
2008–2010	1055	(13.1)	801	(12.4)	
2011–2013	2044	(25.4)	1555	(24.0)	
2014–2016	2638	(32.8)	2288	(35.3)	
2017–2019	1562	(19.4)	1416	(21.8)	
Missing	14	(0.2)	14	(0.2)	
Month, *n* (%)					0.80
January–March	1520	(18.9)	1252	(19.3)	
April–June	2336	(29.0)	1831	(28.2)	
July–September	2289	(28.4)	1836	(28.3)	
October–December	1894	(23.5)	1549	(23.9)	
Missing	14	(0.2)	14	(0.2)	
Days of week, *n* (%)					
Weekdays	5177	(64.3)	4139	(63.9)	0.47
Weekends/Holidays	2876	(35.7)	2342	(36.1)	
Missing	0	(0.0)	1	(<1)	
Call time, *n* (%)					<0.001
Daytime	3589	(44.6)	2617	(40.4)	
Nighttime	4415	(54.8)	3824	(59.0)	
Missing	49	(0.6)	41	(0.6)	
Type of trauma, *n* (%)					<0.001
Traffic accident	4756	(59.1)	4178	(64.5)	
Fall	2150	(26.7)	1425	(22.0)	
Other blunt trauma	992	(12.3)	725	(11.2)	
Penetrating trauma	100	(1.2)	92	(1.4)	
Missing	55	(0.7)	62	(1.0)	
Cause of trauma, *n* (%)					0.22
Accident	7443	(92.4)	5945	(91.7)	
Occupational injury	38	(0.5)	45	(0.7)	
Suicide	344	(4.3)	293	(4.5)	
Violence	94	(1.2)	97	(1.5)	
Others	44	(0.5)	34	(0.5)	
Unknown/Missing	90	(1.1)	68	(1.0)	
**Prehospital vital signs**					
Japan Coma Scale, *n* (%)					<0.001
0	3510	(43.6)	3311	(51.1)	
1–3	1900	(23.6)	1866	(28.8)	
10–30	777	(9.6)	509	(7.9)	
100–300	1430	(17.8)	443	(6.8)	
Unknown/missing	436	(5.4)	353	(5.4)	
Pulse rate: bpm, median (IQR)	96	(81–113)	94	(80–109)	<0.001
Missing, *n* (%)	660	(8.2)	386	(6.0)	
Systolic blood pressure: mmHg, median (IQR)	121	(110–135)	122	(110–136)	0.03
Hypotension	44	(0.5)	33	(0.5)	<0.001
Missing, *n* (%)	1666	(20.7)	771	(11.9)	
Respiratory rate, median (IQR)	24	(20–30)	24	(20–28)	<0.001
Missing, *n* (%)	1335	(16.6)	801	(12.4)	
Cardiopulmonary arrest	346	(4.3)	95	(1.5)	<0.001
**Severity of trauma**					
RTS, median (IQR)	7.84	(6.90–7.84)	7.841	(7.55–7.84)	<0.001
Missing, *n* (%)	1358	(16.9)	907	(14.0)	
AIS					
AIS1 ≥ 3, *n* (%)	2909	(36.1)	1643	(25.3)	<0.001
AIS2 ≥ 3, *n* (%)	74	(0.9)	53	(0.8)	0.51
AIS3 ≥ 3, *n* (%)	18	(0.2)	13	(0.2)	0.77
AIS4 ≥ 3, *n* (%)	1657	(20.6)	1064	(16.4)	<0.001
AIS5 ≥ 3, *n* (%)	450	(5.6)	292	(4.5)	0.003
AIS6 ≥ 3, *n* (%)	266	(3.3)	291	(4.5)	<0.001
AIS7 ≥ 3, *n* (%)	557	(6.9)	552	(8.5)	<0.001
AIS8 ≥ 3, *n* (%)	1149	(14.3)	1182	(18.2)	<0.001
AIS9 ≥ 3, *n* (%)	8	(0.1)	2	(< 0.1)	0.12
ISS, median (IQR)	10	(5–20)	9	(5–16)	<0.001
ISS > 15	2929	(36.4)	1763	(27.2)	<0.001
ISS < 16	4407	(54.7)	4330	(66.8)	
Missing, *n* (%)	717	(8.9)	389	(6.0)	
TRISS, median (IQR)	0.965	(0.870–0.978)	0.968	(0.938–0.988)	<0.001
Missing, *n* (%)	2045	(25.5)	1288	(19.9)	
**Interventions before hospital arrival**					
Oxygen	3970	(49.3)	2963	(45.7)	<0.001
Immobilization	4715	(58.5)	4025	(62.1)	<0.001
Chest compression	300	(3.7)	78	(1.2)	<0.001
Intravenous line placement	87	(1.1)	69	(1.1)	0.93
Defibrillation	13	(0.2)	5	(0.1)	0.15
Intubation	44	(0.5)	23	(0.4)	0.10
Missing	476	(5.9)	350	(5.4)	
**Comorbidities**					
Mental disease	140	(1.7)	132	(2.0)	0.19
Mental retardation	69	(0.9)	80	(1.2)	0.03
Physical disease	24	(0.3)	20	(0.3)	0.91
**On-site time, median (IQR)**	9	(7–11)	19	(16–24)	<0.001

AIS, Abbreviated Injury Scale; IQR, Interquartile range; ISS, Injury Severity Score; RTS, Revised Trauma Score; TRISS, Trauma Injury Severity Score.

**Table 3 children-09-01658-t003:** Multivariable logistic regression analysis with multiple imputation for missing data.

Variables	Odds Ratio	95% Confidence Interval			*p*-Value
Age groups: years					
0			Ref		
1–3	0.78	0.59	to	1.03	0.08
4–6	0.88	0.66	to	1.16	0.35
7–12	1.19	0.92	to	1.56	0.19
13–18	1.53	1.17	to	1.99	0.002
Sex					
Male	0.93	0.86	to	1.01	0.07
Year					
2004–2007			Ref		
2008–2010	1.28	1.09	to	1.49	0.002
2011–2013	1.26	1.09	to	1.46	0.002
2014–2016	1.40	1.22	to	1.61	<0.001
2017–2019	1.47	1.26	to	1.70	<0.001
Month					
January–March			Ref		
April–June	0.96	0.87	to	1.06	0.47
July–September	0.97	0.88	to	1.07	0.53
October–December	1.00	0.90	to	1.11	1.00
Days of week					
Weekdays			Ref		
Weekends/Holidays	1.06	0.99	to	1.14	0.09
Time					
Daytime	0.89	0.83	to	0.96	0.002
Type of trauma					
Traffic accident			Ref		
Fall	0.80	0.73	to	0.88	<0.001
Other blunt trauma	0.69	0.61	to	0.78	<0.001
Penetrating trauma	0.90	0.66	to	1.23	0.52
Cause of trauma					
Accident			Ref		
Occupational injury	1.37	0.88	to	2.14	0.17
Suicide	1.27	1.03	to	1.57	0.03
Violence	1.74	1.27	to	2.38	0.001
Others	1.11	0.70	to	1.76	0.67
**Prehospital vital signs**					
Hypotension	1.52	0.92	to	2.50	0.10
**Severity of trauma**					
RTS	1.251	1.194	to	1.309	<0.001
AIS					
AIS1 ≥ 3	0.77	0.71	to	0.84	<0.001
AIS2 ≥ 3	0.97	0.67	to	1.41	0.89
AIS3 ≥ 3	1.05	0.49	to	2.25	0.90
AIS4 ≥ 3	0.84	0.76	to	0.92	<0.001
AIS5 ≥ 3	0.84	0.72	to	0.99	0.04
AIS6 ≥3	1.25	1.04	to	1.50	0.02
AIS7 ≥ 3	1.26	1.11	to	1.44	0.001
AIS8 ≥ 3	1.25	1.14	to	1.37	<0.001
AIS9 ≥ 3	0.50	0.10	to	2.47	0.39
**Interventions before hospital arrival**					
Oxygen	0.85	0.78	to	0.93	<0.001
Immobilization	1.16	1.06	to	1.27	0.001
Chest compression	1.15	0.75	to	1.75	0.52
Intravenous line placement	1.41	1.00	to	2.01	0.05
Defibrillation	1.05	0.32	to	3.43	0.94
Intubation	1.73	0.99	to	3.03	0.05
**Comorbidities**					
Mental disease	0.94	0.71	to	1.24	0.67
Mental retardation	1.56	1.11	to	2.18	0.009
Physical disease	1.02	0.55	to	1.88	0.96

AIS, Abbreviated Injury Scale; RTS, Revised Trauma Score.

**Table 4 children-09-01658-t004:** Multivariable logistic regression analysis with complete data.

Variables	Odds Ratio	95% Confidence Interval			*p*-Value
Age groups: years					
0			Ref		
1–3	0.64	0.34	to	1.22	0.18
4–6	0.56	0.30	to	1.04	0.07
7–12	0.75	0.41	to	1.39	0.37
13–18	0.99	0.54	to	1.83	0.98
Sex					
Male	0.88	0.80	to	0.96	0.007
Year					
2004–2007			Ref		
2008–2010	1.30	1.08	to	1.56	0.005
2011–2013	1.38	1.16	to	1.63	<0.001
2014–2016	1.48	1.25	to	1.75	<0.001
2017–2019	1.46	1.23	to	1.74	<0.001
Month					
January–March			Ref		
April–June	0.93	0.82	to	1.05	0.23
July–September	0.92	0.82	to	1.04	0.21
October–December	0.98	0.86	to	1.11	0.71
Days of week					
Weekdays			Ref		
Weekends/Holidays	1.10	1.00	to	1.20	0.04
Time					
Daytime	0.91	0.83	to	0.99	0.03
Type of trauma					
Traffic accident			Ref		
Fall	0.89	0.79	to	1.01	0.06
Other blunt trauma	0.63	0.54	to	0.74	<0.001
Penetrating trauma	1.04	0.67	to	1.61	0.86
Cause of trauma					
Accident			Ref		
Occupational injury	1.19	0.73	to	1.94	0.48
Suicide	1.06	0.81	to	1.37	0.68
Violence	1.70	1.14	to	2.53	0.009
Others	1.18	0.65	to	2.13	0.59
**Prehospital vital signs**					
Hypotension	1.42	0.77	to	2.59	0.259
**Severity of Trauma**					
RTS	1.270	1.197	to	1.348	<0.001
AIS					
AIS1 ≥ 3	0.76	0.69	to	0.85	<0.001
AIS2 ≥ 3	0.79	0.51	to	1.24	0.31
AIS3 ≥ 3	1.14	0.38	to	3.40	0.81
AIS4 ≥ 3	0.85	0.76	to	0.95	0.004
AIS5 ≥ 3	0.86	0.72	to	1.04	0.13
AIS6 ≥ 3	1.48	1.20	to	1.83	<0.001
AIS7 ≥ 3	1.26	1.07	to	1.49	0.007
AIS8 ≥ 3	1.22	1.09	to	1.37	0.001
AIS9 ≥ 3	1.28	0.10	to	15.56	0.85
**Interventions before hospital arrival**					
Oxygen	0.81	0.73	to	0.90	<0.001
Immobilization	1.07	0.96	to	1.19	0.21
Chest compression	1.96	0.81	to	4.72	0.13
Intravenous line placement	1.20	0.74	to	1.93	0.46
Intubation	1.78	0.65	to	4.85	0.26
**Comorbidities**					
Mental disease	0.89	0.64	to	1.24	0.49
Mental retardation	1.52	1.04	to	2.23	0.03
Physical disease	0.94	0.46	to	1.93	0.87

AIS, Abbreviated Injury Scale; RTS, Revised Trauma Score.

## Data Availability

The authors are not authorized to distribute the data sets used in this study. Data was obtained from the Japan Trauma Data Bank managed by Japanese Association for the Surgery of Trauma (Trauma Surgery Committee) and the Japanese Association for Acute Medicine (Committee for Clinical Care Evaluation) and are available at https://www.jtcr-jatec.org/traumabank/index.htm with the permission of the Japan Trauma Data Bank (accessed on 25 October 2022).

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
