# Peer review of "Factors Associated with an Increase in On-Site Time of Pediatric Trauma Patients in a Prehospital Setting: A Nationwide Observational Study in Japan"

_children, 2022, doi:10.3390/children9111658_

Round 1
Reviewer 1 Report
While this is a good study specifically for the medical system in Japan, it is hard to generalize to other systems. However, it can definitely lead to quality improvement projects in Japan. I did have concerns on how they define hypotension. How did they obtain the definitions. Also, I have questions regarding the statistical methods. For the univariate analysis, how is just 2 points in bpm of the heart rate statistically significant? Also, if the same percentage in the two groups are hypotensive, how is that also statistically significant? Is it taking into account the missing data?
Also, just a grammatical issue, I think line 50 is probably "not known" instead of "now known"
Author Response
Response to reviewer 1 comment
While this is a good study specifically for the medical system in Japan, it is hard to generalize to other systems. However, it can definitely lead to quality improvement projects in Japan.
1.I did have concerns on how they define hypotension. How did they obtain the definitions?
Response
Thank you for your comments. The database contained the systolic blood pressure on-site. We defined hypotension using systolic blood pressure. We showed the definition and the relevant referenceï¼»28ï¼½.
2.Also, I have questions regarding the statistical methods. For the univariate analysis, how is just 2 points in bpm of the heart rate statistically significant?
Response
Thank you for your comments. We calculated again. It is statistically correct, however, due to the large number of patients involved, it can be said that there is no difference between the two groups clinically, although statistically there is a difference.
3.Also, if the same percentage in the two groups are hypotensive, how is that also statistically significant? Is it taking into account the missing data?
Response
Thank you for your comments. As the reviewer says, it is taking into account the missing data. Furthermore, due to the large number of patients involved, even the small difference showed statistical significance.
4.Also, just a grammatical issue, I think line 50 is probably "not known" instead of "now known"
Response
Thank you for your comments. We corrected the words.
Changes
now known→not known
Thank you for reviewing our manuscript.
Reviewer 2 Report
The article was very interesting and the findings (need for more doctors) logical and well back by the data collected. There was maybe a little too many tables (however this was a qualitative study) therefore it could be excused. Maybe a little more description regarding the findings as all the abbreviations were rather bothersome when reading the results and conclusions.
Generally well done and good scientific methodology with sound statistics.
Author Response
Response to reviewer 2
The article was very interesting and the findings (need for more doctors) logical and well back by the data collected. There was maybe a little too many tables (however this was a qualitative study) therefore it could be excused. Maybe a little more description regarding the findings as all the abbreviations were rather bothersome when reading the results and conclusions.
Generally well done and good scientific methodology with sound statistics.
Response
We discussed the revisions.
Regarding the reduction of the charts, we found it difficult for the scientific explanation of the study. Furthermore, we have considered spelling out some of the abbreviations (e.g. RTS) in the manuscript as a solution. However, it could be rather bothersome. We are very sorry, but we left the abbreviations as they are.
Thank you for your comments.